# THE ADVANTAGE OF USING STUDENT'S T-PRIORS IN VARIATIONAL AUTOENCODERS

## ABSTRACT

Is it optimal to use the standard Gaussian prior in variational autoencoders? With Gaussian distributions, which are not weakly informative priors, variational autoencoders struggle to reconstruct the actual data. We provide numerical evidence that encourages using Student's t-distributions as default priors in variational autoencoders, and we challenge the usual setup for the variational autoencoder structure by comparing Gaussian and Student's t-distribution priors with different forms of the covariance matrix.

## 1 INTRODUCTION

Variational autoencoders (VAEs) (Kingma & Welling, 2013; Rezende et al., 2014) are generative models that, by learning a joint distribution on all variables, aim to reveal the process of how datasets are generated. Consider input data $\mathbf{X} = \{\mathbf{x}_1, \ldots, \mathbf{x}_n\}$ where $\mathbf{x} \in \mathcal{R}^d$ and $d$ is the feature dimension, and $k$-dimensional latent variables $\mathbf{z}$. VAEs consist of two neural networks, inference and generative networks (encoder $q_\phi(\mathbf{z}|\mathbf{x})$ and decoder $p_\theta(\mathbf{x}|\mathbf{z})$) that are jointly trained to maximize a variational lower bound log likelihood of the data,

$$\mathcal{L}(\theta, \phi; \mathbf{x}, \mathbf{z}) = \mathop{\mathbb{E}}_{q_\phi(\mathbf{z}|\mathbf{x})} [\log(p_\theta(\mathbf{x}|\mathbf{z}))] - D_{KL}(q_\phi(\mathbf{z}|\mathbf{x})\|p_\theta(\mathbf{z})). \tag{1}$$

The KL divergence term acts as the regularizer that minimizes the distance between $q_\phi(\mathbf{z}|\mathbf{x})$ (approximated posterior) and the suggested prior, $p(\mathbf{z})$.

Despite the recent successes of VAEs (Kingma et al., 2014; Gregor et al., 2015; Kulkarni et al., 2015; Hsu et al., 2017), there is still room for improvement concerning the approximation of the true posterior distribution. The main difficulty in VAE is the "posterior collapse", in which the term $q_\phi(\mathbf{z}|\mathbf{x})$ stays in the initial state, equivalent to random samples from the prior (Bowman et al., 2015; Chen et al., 2016; Kingma et al., 2016) that results in uninformative latent variables. Various extensions to VAE models have been proposed to tackle this problem, such as weakening the decoder (Bowman et al., 2015; Chen et al., 2016; Sønderby et al., 2016) or by changing the objective of VAEs (Zhao et al., 2017; Higgins et al., 2017). Many of these studies aim to learn a more expressive posterior, which results in a tighter variational bound.

As a solution to the problem of the posterior collapse, we will explore different prior distributions for the VAE, specifically various forms of prior's covariance matrices. The standard prior, in connection with VAE, is the multivariate Gaussian with a diagonal covariance, $q(\mathbf{z}|\mathbf{x}) = \mathcal{N}(z|\mu(\mathbf{x}), diag(\sigma^2(\mathbf{x})))$. Using Gaussian diagonal covariance results in Mean-Field Variational Inference (MFVI). MFVI is an approximation method that was adopted from mean-field theory (Opper & Saad, 2001) and introduced by (Peterson, 1987) when training neural networks. It considers a fully factorized variational distribution, in the VAE case $q_\phi(\mathbf{z}) = \prod_i^d q_\phi(\mathbf{z}_i)$, that is sufficient to approximate the posterior and provide a tractable approximation ($d$ being the size of the latent space). However, these models may lead to the poor hidden representation since they ignore all correlation between latent variables (Hoffman & Johnson, 2016).

Recently, it has shown how to use the Student's t-distribution prior when training VAE's and that it has a positive effect on the reconstruction ability (Abiri & Ohlsson, 2019). The covariance matrix used for that study was limited to a diagonal form. Here, we examine the effect of the covariance in priors by extending the VAE to allow for full covariance in both Gaussian and Student's t families of priors. The models are tested on two different datasets with the intent to highlight the advantages of

using a prior different from the standard diagonal Gaussian. Using the Student's t-distribution prior, the degree of freedom parameter can be shared between the prior and the posterior. Hence, we have an empirical Bayes method that provides the ability to update the prior.

## 2 METHODS

We investigate two possible families of priors for VAE, Gaussian, and Student's t. All priors in this work are standardized, i.e location zero and scale one. For the Gaussian family, besides the centered isotropic multivariate Gaussian, we also analyze the effect of multivariate Gaussian (MVN) with full-covariance on the model (both forms have been mentioned in the original VAE paper (Kingma & Welling, 2013)).

To set up the network to use full-covariance MVN, instead of optimizing the diagonal vector $\sigma$ with dimension $d$, we need to optimize the covariance matrix $\Sigma$ with the dimension $(d \times d)$. Since $\Sigma$ is positive semi-definite we can use the Cholesky decomposition, $\Sigma = LL^T$, to only optimize the lower (or upper) triangular matrix $L$ with the dimension of $\frac{d(d+1)}{2}$. The KL divergence of an MVN and a normal prior is,

$$D_{KL}(q_\phi(\mathbf{z}|\mathbf{x})\|p_\theta(\mathbf{z})) = \frac{1}{2}\left\{-\log(|\Sigma|) + \text{tr}(\Sigma) + \mu^T\mu - d\right\}, \quad (2)$$

where $\mu$ is the location of the distribution $q_\phi(\mathbf{z}|\mathbf{x})$. This expression allows for a straightforward implementation where the full covariance matrix $\Sigma$ can be updated using standard stochastic gradient descent techniques. Using the full covariance prior increases the number of parameters, and it grows as $d^2$, compared to a linear growth for the standard diagonal Gaussian prior.

The next family is the Student's t-distribution. It has been shown that (Abiri & Ohlsson, 2019), with the use of implicit differentiation of the cumulative function of the Gamma distribution (Figurnov et al., 2018), it is possible to use a Student's t-distribution, which belongs to the location-scale family, as a prior for the VAE. In this work, we considered the case of multivariate t-distributions (MVT) with a diagonal covariance, in which all the marginal distributions share the same degree of freedom parameter $\nu$. In this case, there is an analytical solution for the KL divergence,

$$D_{KL}(q_\phi(\mathbf{z}|\mathbf{x})\|p_\theta(\mathbf{z})) = -\log(|\tilde{\Sigma}|) - \left(\frac{\nu+d}{2}\right)\left(\psi(\frac{\nu+d}{2}) - \psi(\frac{\nu}{2}) + \mathbb{E}_{q_\phi}\left[\log(1 + \frac{z^2}{\nu})\right]\right), \quad (3)$$

with $\psi(\cdot)$ being the digamma function, $d$ the latent dimension and $\tilde{\Sigma}$ the diagonal covariance matrix. The expectation in equation 3 can be approximated by a numerical integration method (Abiri & Ohlsson, 2019). Unlike Gaussian random variables where a diagonal covariance matrix cause independency, uncorrelated Student's t random variables are not in general independent (Kotz & Nadarajah, 2004). This can be illustrated looking at the joint PDF of two Student's t random variables $z_1$ and $z_2$,

$$p(z_1, z_2) = \text{St}\left(\begin{bmatrix} z_1 \\ z_2 \end{bmatrix}; \begin{bmatrix} \mu_1 \\ \mu_2 \end{bmatrix}, \begin{bmatrix} \Sigma_{11} & 0 \\ 0 & \Sigma_{22} \end{bmatrix}, \nu\right)$$

$$\propto \left(1 + \frac{1}{\nu}\sum_{i=1,2}(z_i - \mu_i)^T\Sigma_{ii}^{-1}(z_i - \mu_i)\right)^{-\frac{\nu+p}{2}}$$

which cannot factorize into a product of marginal densities. Similar to MVN, MVT can be extended to full covariance matrix by merely replacing $\tilde{\Sigma}$ in equation 3 by the full covariance matrix $\Sigma$.

The last prior we consider in our study is a batch of Student's t-distributions, obtained by stacking several independent t-distributions to form the latent space, each with separate parameters including the degree of freedom $\nu$. Since the joint probability of such a batch does not have an analytical density function, we use Monte Carlo sampling to estimate the KL divergence term,

$$\log p(\mathbf{z}) + \log q(\mathbf{z}|\mathbf{x})$$

In this study we used a single MC sample for this estimation.

Figure 1 shows an illustration of the five different VAE priors, where a simple VAE with two inputs are connected directly to a $z$-layer of size two. Each vertical row of $(\mu, \Sigma, \nu)$ parameters corresponds to one of the five priors. Table 1 shows a condensed summary of all five priors. From now on, we refer to them by their abbreviations (shown by bold text).

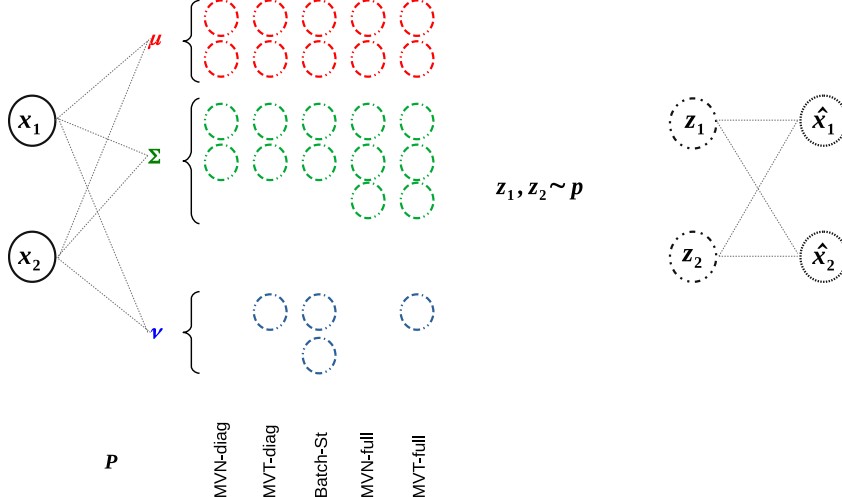

Figure 1: An illustrative comparison of all five priors for a simple 2D input VAE. The latent dimension is two and the priors are parameterized using $(\mu, \Sigma, \nu)$, as indicated by the vertical rows. For the full covariance models $\Sigma$ has three parameters that form a lower/upper triangular positive semi-definite matrix. All the marginals of MVT-diag and MVT-full share one $\nu$, the Batch-St prior, on the other hand, has two independent $\nu$-parameters.

Table 1: Five different priors for VAE with their properties. For the posterior dimension, $d$ refers to the size of the latent space.

| VAE priors | | | |
|---|---|---|---|
| Name | Posterior | Posterior dimension | KL divergence |
| **MVN-diag** | MVN with diagonal covariance | $2d$ | Analytically available |
| **MVN-full** | MVN with full covariance | $d(1 + \frac{d+3}{2})$ | Analytically available |
| **MVT-diag** | MVT diagonal covariance | $2d + 1$ | Analytically available |
| **MVT-full** | MVT full covariance | $d(1 + \frac{d+3}{2}) + 1$ | Analytically available |
| **Batch-St** | Batch of Student's t | $3d$ | sample MC |

## 3 EXPERIMENTS AND DISCUSSIONS

We evaluated the five VAE priors with their specific structures on two datasets:

- Gaussian ovals: A 2D data set that contains 1500 samples generated data from 5 bivariate normal distributions with diagonal covariances. The "clusters" are different in both shape and width.

- Omniglot (Lake et al., 2015): This data with overal dimensions of $(32460, 105 \times 105)$, contains characters from 50 different alphabets. 20 random handwritten samples represent each character. In the original work, the data were divided into two groups of 30 alphabets for training and 20 for testing purposes. In our study, we used the same division into training and test.

For the Gaussian ovals, we used root mean squared error (RMSE), and for Omniglot data, we used the structural similarity index measure (SSIM) (Wang et al., 2004). Unlike other image difference measurements, SSIM measures the perceptual difference between two similar images. The final SSIM here is the average over all images differences.

### 3.1 GAUSSIAN OVALS

The Gaussian ovals is a 2D dataset where each sample $(x_1, x_2)$ is generated from one of five Gaussian distributions with randomly picked scale and location parameters. The dataset contains 1500

samples. For the experiments, repeated cross-validation was used, with approximately 70% for training and the remaining for the test. Figure 2 (left plots) shows one instance of the Gaussian ovals test set.

Since $x_1$ and $x_2$ are independent, a simple VAE network with a latent layer of dimension two and no hidden layers should be able to reconstruct the input with a small RMSE. A VAE model with this structure and MVN-diag as prior will end up with only 18 parameters (with bias nodes). To see the effect of using different values for the standard deviation of the MVN-diag prior, we trained four VAE models with standard deviations of $[1., 0.5, 0.1, 0.01]$. Each model was evaluated on the test data 10 times with a random split of training and test data.

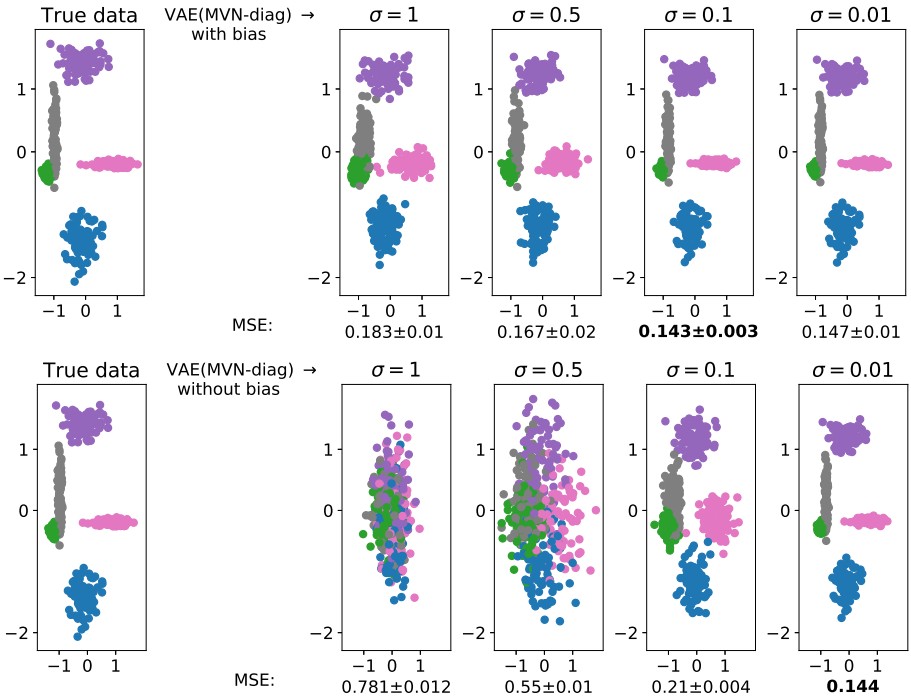

Figure 2: Best viewed in color (color-coded). The leftmost plots show the test data (denoted True data) and the plots on the right show reconstructed test data using different values for $\sigma$. **Upper row:** Results from the VAE model with bias nodes, **Lower row:** Result from the VAE model without bias nodes (in total 12 parameters). The presented MSE reconstruction errors are estimated using cross-validation, repeated 10 times.

Figure 2 shows one test set reconstruction of the input with the different $\sigma$ used when training the VAE (MVN-diag) model. The upper row of the figure shows reconstruction results for the VAE model with bias nodes, whereas in the lower row, all bias nodes were removed from the model. For both cases (with and without bias nodes), decreasing standard deviation improves the reconstruction and separation of the ovals. In this case, the VAE tends to a deterministic autoencoder and learns a manifold. Since some of the ovals have been generated from Gaussian with a small standard deviation, a normal prior with $\sigma = 1$ is too informative and results in poor network reconstruction. This is most apparent for the models without bias nodes. The MVN-diag prior, apart from being too informative, may need tuning of the $\sigma$ parameter to increase performance. In practice, the default $\sigma = 1$ is probably most used. Using a weakly informative prior, such as the Student's t-distribution, will improve the reconstruction results without any parameter tuning for the prior.

For the next set of experiments on the Gaussian ovals, all five VAE priors from Table 1 were tested. All VAE models were trained without any bias nodes, and from now on the prior's standard deviation was fixed to 1 to compare the effect of using different prior distributions. Again MSE reconstruction errors were estimated using ten random training and test data splits.

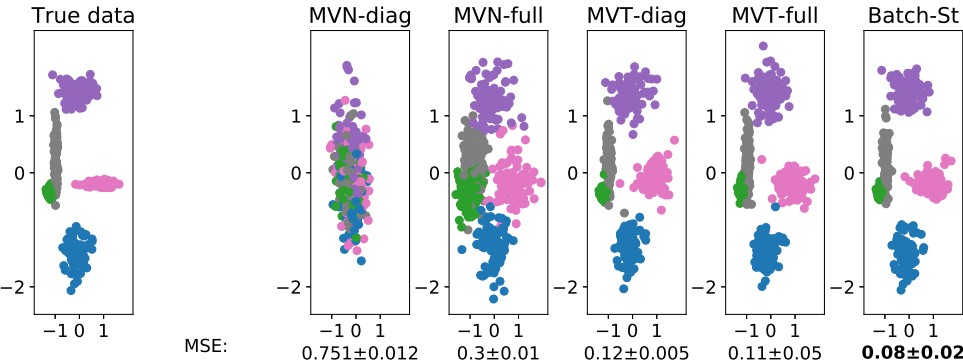

Figure 3: The leftmost plot shows the test data (denoted True data), and the plots to the right show reconstructed test data using VAE models with five different priors. All models were implemented without any bias nodes. The presented MSE reconstruction errors are estimated using cross-validation, repeated 10 times.

Figure 3 shows reconstructed test data (one instance) for all five prior distributions. The smallest average MSE was found for the batch of Student-t as priors (Batch-St). Since the real variance of Batch-St for each $z_i$ is $\sigma_i\sqrt{\frac{\nu_i}{\nu_i-2}}$ and $\nu_i$, the degree of freedom for each of the Student's t is shared between the prior and posterior, the model can update its prior during the optimization. This is also the case for MVT-diag and MVT-full, except that all the marginal distributions share the same $\nu$, only a single parameter updating in the prior, which is less powerful than Batch-St model. Although the reconstruction improved by using full covariance matrix for the Gaussian prior (MVN-full) compared to the standard MVN-diag prior, both Gaussian priors perform worse compared to Student's t priors. We have done a similar analysis with the presence of bias nodes in all of the VAE models. With bias nodes, the error drops for all models with an RMSE $\approx 0.02$. By observing the posterior learned parameters, one can conclude that the two Gaussian priors stay close to the initial setting, while the Student's t priors do learn even when bias nodes are present.

Finally, it is intuitive to observe the distribution of latent space for all models. Figure 4 shows the priors (top row) and the learned marginal distribution $q(\mathbf{z})$ (bottom row) after training. All models, except MVN-diag, allow for a better fit between the posterior and the prior and improve the flexibility of $q_\phi(\mathbf{z}|\mathbf{x})$, which results in a better reconstruction that resembles the underlying input data clusters.

## 3.2 Omniglot data

For the Omniglot data, we used the proposed fixed division of training and test data. The training data contains 19280 samples (characters) from one group of alphabets, and the test data contains 13180 samples from another separate group of alphabets. This experiment aimed to study the effect of using the five different VAE prior distributions for a more complicated image data set. The VAE structure still contained a very simple encoder and decoder part to put the focus on learning an appropriate posterior distribution. Hence, the 2D input layer was directly connected to the latent layer, with a selected size of 25, which was then directly connected to the 2D output layer. In case of, e.g., the Student's t diagonal (MVT-diag) prior, this resulted in a latent layer with 51 parameters, 25 for each of the location ($\mu$) and scale ($\Sigma$) parameters and one common degree of freedom ($\nu$). No bias nodes were used in the VAE models. All models were trained with the same set of common hyperparameters, mini-batch size of 32, learning rate $10^{-4}$ and the number of epochs equal to 1000.

Figure 5 (left plot) shows examples of four different characters: the original image and VAE generated images for the different priors. Visually, the VAE with Gaussian diagonal prior (MVN-diag) produces more blurry images compared to the other prior distributions. This can also be seen by comparing test set average SSIM values, where the MVN-diag prior is clearly generating images with lower SSIM. Even though the difference between e.g., Batch-St and MVN-full is small, 0.793

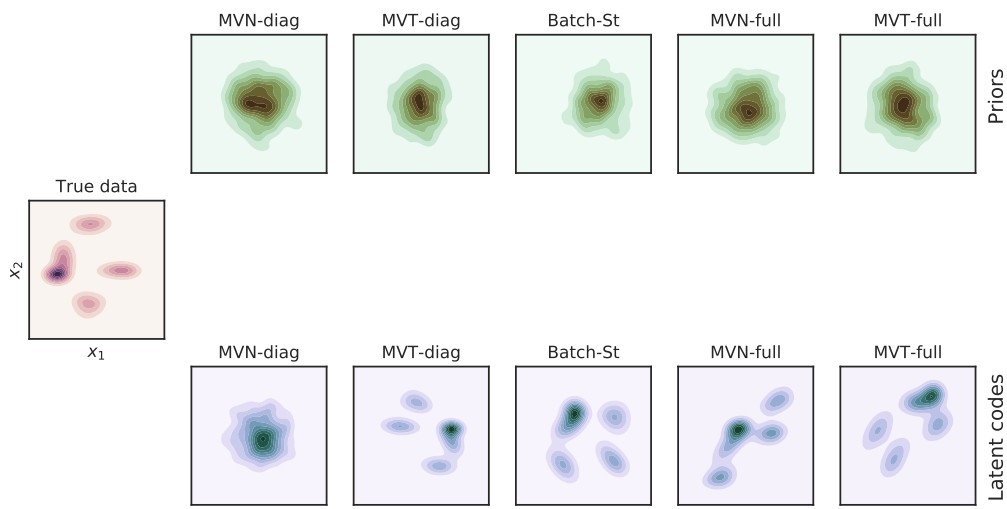

Figure 4: This figure shows the priors and posteriors of latent variable of all five models. **Lower left** shows the distribution of the data. The **top row** contains all priors that have been used for the models, and the **second row** shows the marginal distribution $q(\mathbf{z})$ of the models encoder after training.

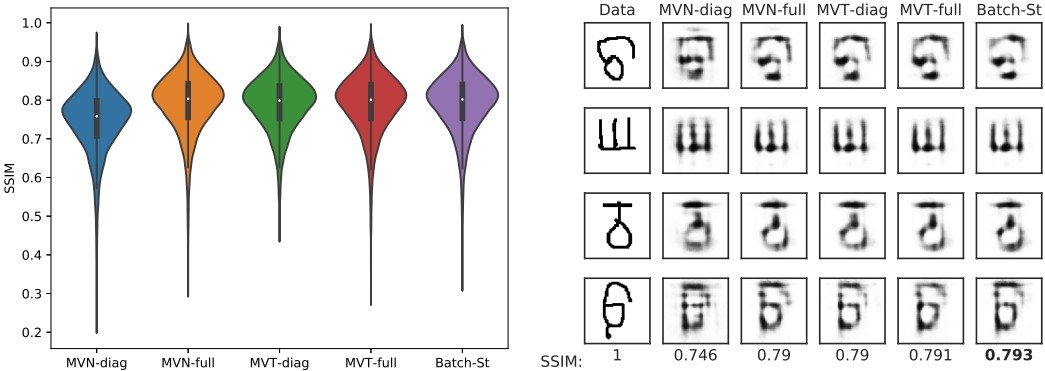

Figure 5: **Left** figure shows distributions of SSIM (higher is better) for each method on x-axis. **Right** figure shows reconstructed samples from four random characters from test data. The numbers on the last row are the SSIM between the original character and the reconstructed one.

compared to 0.790, a paired Mann-Whitney U test shows a significant difference in favor of the Batch-St prior. Note that in this experiment, we did not optimize the VAE architecture for optimal SSIM between original and generated images. The purpose was to show the advantage of moving away from the MVN-diag prior, where either MVT-diag or Batch-St prior distributions are likely to be better.

One consequence of using the full covariance priors is the increase of parameters. We have timed the training and testing process for all models on the Omniglot data to quantify the increase in computational cost when adding more parameters. All calculations were carried out on a machine with a single NVIDIA GV100 (TITAN V) graphics card.

Table 2: Training and testing times for the all five VAE models using the Omniglot data.

| Methods | MVN-diag | MVN-full | MVT-diag | MVT-full | Batch-St |
|---|---|---|---|---|---|
| Time (second) | 1220 | 2070 | 1240 | 2200 | 1730 |

The result in Table 2 shows the approximate training and testing time for each method. As expected, the full covariance models requires more computational time due to more parameters in the latent layer. The Batch-St prior requires less parameters compared to MVN-full and MVT-full, but the computational cost depends heavily on the MC sampling of KL divergence. Here we sample once, making Batch-St faster compared to the full covariance models.

## 4 CONCLUSION

We have investigated the effect of using different prior distributions for the variational autoencoder. Besides the standard diagonal Gaussian prior (with location zero and scale one), we have used a Gaussian prior with full covariance, as well as the Student's t prior, both the diagonal and full covariance structure. Finally, we tested a batch of independent Student's priors. We have deliberately used very simple VAE structures to focus on the effect of different priors in latent variable learning.

By comparing the two Gaussian priors, we observed a clear advantage of using the full covariance structure, with the drawback of requiring more parameters. Using a weakly informative prior such as the Student's t with a lower number of parameters compared to a more informative full covariance Gaussian is advantageous for VAE. Furthermore, our setup of the Student's t-priors allows VAE to update the prior's scale parameters, providing an empirical Bayes model and results in a more flexible posterior approximation. All three Student's t-priors show similar performance on the test data, with a slight advantage of using batches of Student's t-priors (Batch-St). This distribution, however, has the drawback of needing Monte Carlo estimates of the KL divergence term due to there being no analytical form of the density function. The diagonal Student's t-prior distribution is in this respect a better alternative, also requiring fewer parameters.

Regardless of the exact choice of Student's t-prior, the main message is that the standard diagonal Gaussian prior is far from optimal for the variational autoencoder. It is highly recommended to use a Student's t-distribution prior.

ACKNOWLEDGMENTS

The authors would like to thank Professor Carsten Peterson and Pontus Fyhr for valuable discussions and comments.

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
