# OpenReview forum: "The advantage of using Student's t-priors in variational autoencoders"
_ICLR.cc/2020/Conference — Reject_

### Official Review · AnonReviewer1 · 2019-10-21
**Official Blind Review #1**

**Rating:** 1

**Review:**

Summary:  This paper studies alternative priors for VAEs, comparing Normal distributions (diagonal and full covariance) against Student-t’s (diagonal and full covariance).   In particular, the paper is concerned with posterior collapse---i.e. posterior remains at the prior, limiting the model’s ability to reconstruct the data.  Experiments are performed on a synthetic 2D dataset, ‘Gaussian ovals,’ and on OMNIGLOT.  Results primarily take the form of visualizations of the reconstructed data and MSE / SSIM numbers.

Pros:  Systematic study and investigation of alternative priors for deep generative models is an under-studied area.  Moreover, heavy-tailed priors such as the student-t---while widely successful for robust regression---have not been explored as extensively for latent variable models, to the best of my knowledge.  This paper makes some steps towards solving these open problems.

Cons:  I have two primary critiques of the paper: (i) the experimental hypothesis is unclear, (ii) no engagement with the results of Mathieu et al. [ICML 2019], who also study Student-t priors for reconstruction (and disentanglement).

Regarding (i), the paper seems to be testing two hypotheses simultaneously: the effect of diagonal vs full covariance matrices and exponential-tailed vs heavy-tailed priors.  The latter seems more crucial for purposes of reconstruction according to Figure 3 (since the diagonal St-t has good reconstruction).  Yet a proper study of the effect of tails vs reconstruction would report results as the degree of freedom parameter is gradually increased (as this controls the tails directly).  No careful ablation study of this sort is performed.  For comparison, see Figure 2 of Mathieu et al. [ICML 2019].  Moreover, the text simply calls the student-t a “weakly informative prior,” but it needs to be much more specific about what characteristics of the student-t are crucial.  If all we require is something “weakly informative,” why aren’t alternatives like a diffuse Gaussian or uniform also considered?

Regarding (ii), Mathieu et al. [ICML 2019] also study priors formed by products of student-t marginals, but their work is not cited.  Mathieu et al. [ICML 2019] also show that, due to student-t’s not being rotationally invariant (unlike the diagonal Gaussian), they improve disentanglement with only a minor degradation of reconstruction.  As this work also studies reconstruction in student-t VAEs, it should include some discussion of Mathieu et al. [ICML 2019]’s results---if not direct engagement with their hypotheses.

Final Evaluation:  While I like the general motivation for this work, there are no clear experimental hypotheses being tested in the experiments.

Mathieu, E., Rainforth, T., Narayanaswamy, S. and Teh, Y.W., 2018. Disentangling Disentanglement in Variational Autoencoders. ICML 2019.

**Experience Assessment:**

I have published in this field for several years.

**Review Assessment: Checking Correctness Of Derivations And Theory:**

I assessed the sensibility of the derivations and theory.

**Review Assessment: Checking Correctness Of Experiments:**

I assessed the sensibility of the experiments.

**Review Assessment: Thoroughness In Paper Reading:**

I read the paper thoroughly.

---

### Official Review · AnonReviewer3 · 2019-10-22
**Official Blind Review #3**

**Rating:** 1

**Review:**

The paper proposes using more expressive priors in variational autoencoders (VAEs), such as the multivariate Student’s t-distribution. The experiments demonstrate a small improvement of SSIM metric on Omniglot.

Exploring various design choices of the VAE, including the prior distribution, is important, However, I feel that the paper does not have a sufficient contribution. The theoretical novelty is small; the experimental evaluation is lacking and missing crucial baselines, such as normalizing flows. The text is also not very well-written. Because of these factors, I think the paper should be rejected.

In more detail, I see the following issues with the current manuscript:
1. The manuscript is not properly anonymized as it includes acknowledgements.
2. The title of the paper is confusing, as it suggests that only the prior is being changed. However, Table 1 shows that the posterior is changed jointly with the prior.
3. The novelty compared to (Abiri & Ohlsson, 2019) is very small. The difference is replacing diagonal covariance multivariate Student’s t-distribution with similar distributions, such as the full covariance Student’s t-distribution or a batch of Student’s t-distributions. The empirical improvement from these changes, based on Figure 5, is quite small.
4. The text of the paper can be significantly improved. There are many typos, e.g. “lower bound log likelihood” instead of “lower bound on the log-likelihood”. There are also issues in the math, such as a missing negation in the KL divergence at the bottom of Page 2. Some things are not specified: how exactly the diagonal values of the Cholesky decomposition are enforced to be positive? what are the bias nodes mentioned in Figure 2? what is the multivariate Student’s t-distribution density function?
5. One thing I didn’t understand is why the KL-divergence of a product of independent (“batch”) t-distributions cannot be computed analytically using equation (3), given that the KL(prod_i q_i || prod_i p_i) factorizes as \sum_i KL(q_i || p_i).
6. The experimental protocol is lacking. A key missing comparison is normalizing flows (https://arxiv.org/abs/1606.04934) which are typically used to make the VAE prior distributions more expressive. There are also VampPrior (https://arxiv.org/abs/1705.07120) and LARS (https://arxiv.org/abs/1810.11428) which also augment VAE’s prior in various ways. Another issue is that only the SSIM metric is reported, whereas the standard way of comparing generative models is the test log-likelihood.

**Experience Assessment:**

I have published one or two papers in this area.

**Review Assessment: Checking Correctness Of Derivations And Theory:**

I assessed the sensibility of the derivations and theory.

**Review Assessment: Checking Correctness Of Experiments:**

I assessed the sensibility of the experiments.

**Review Assessment: Thoroughness In Paper Reading:**

I made a quick assessment of this paper.

---

### Official Review · AnonReviewer2 · 2019-10-23
**Official Blind Review #2**

**Rating:** 1

**Review:**

This paper advertised the use of student-t prior, instead of the standard normal prior, in variational autoencoders. The authors listed a few candidates, including the diagonal and full covariance matrices. Several experiments on two datasets were then performed.

In general, this paper is confusing and difficult to follow. For example, the same terminology has different meanings across this paper. The novelty in this paper is also quite limited, as the main technical part is basically to list Equations (2) and (3) to show the KL divergence for normal and student-t distributions. Finally, the experiments need to be significantly improved, as there are not sufficient takeaways.
1. The authors used the word "prior" to denote different distributions, e.g., p (z) and q (z | x). This is pretty confusing, as q (z | x) is usually treated as the posterior. In Table 1, the posterior was even described under VAE priors. It would be better for the authors to read the original VAE papers (e.g., Kingma & Welling, ICLR 2014) carefully to understand these terminologies: https://arxiv.org/pdf/1312.6114.pdf
2. The contributions of this paper are far from sufficient: In Section 2, the main takeaway for me is the list of KL divergence from different distributions, which is pretty standard and difficult to be treated as the authors' own contributions.
3. I feel confused about the purposes of Figures 2 and 3. First, they were solely focused on reconstruction tasks, which may not be interesting enough, especially on such a toy dataset. Furthermore, the standard deviations in Figure 3 were set to be one, which may not be fair for MVN-Diag, as Figure 2 suggests that one is a bad choice. As a result, it's not convincing to conclude that student-t is the best candidate.
4. Again, the "prior" used at the top row of Figure 4 is difficult to understand. How did you compute the scale parameter and plot the figure?

**Experience Assessment:**

I have published one or two papers in this area.

**Review Assessment: Checking Correctness Of Derivations And Theory:**

N/A

**Review Assessment: Checking Correctness Of Experiments:**

I carefully checked the experiments.

**Review Assessment: Thoroughness In Paper Reading:**

I read the paper thoroughly.

---

### Decision · Program_Chairs · 2019-12-19

**Decision:**

Reject

**Comment:**

The consensus among all reviewers was to reject this paper, and the authors did not provide a rebuttal.